# Radiation-Induced Hypothyroidism in Patients with Oropharyngeal Cancer Treated with IMRT: Independent and External Validation of Five Normal Tissue Complication Probability Models

**DOI:** 10.3390/cancers12092716

**Published:** 2020-09-22

**Authors:** Zuzanna Nowicka, Bartłomiej Tomasik, Anna Papis-Ubych, Robert Bibik, Łukasz Graczyk, Tomasz Latusek, Tomasz Rutkowski, Krystyna Wyka, Jacek Fijuth, Jonathan D. Schoenfeld, Justyna Chałubińska-Fendler, Wojciech Fendler

**Affiliations:** 1Department of Biostatistics and Translational Medicine, Medical University of Lodz, 92-215 Lodz, Poland; zuzanna.nowicka@stud.umed.lodz.pl (Z.N.); bartlomiej.tomasik@umed.lodz.pl (B.T.); 2Department of Radiotherapy, Medical University of Lodz, 95-513 Lodz, Poland; jacek.fijuth@umed.lodz.pl; 3N. Copernicus Memorial Regional Specialist Hospital, Department of Radiotherapy, 95-513 Lodz, Poland; anna_papis@interia.pl; 4Department of Radiation Oncology, Oncology Center of Radom, 26-600 Radom, Poland; rmbibik@wp.pl (R.B.); L.GRACZYK@ONKOLOGIARADOM.PL (Ł.G.); 5Radiotherapy Department, Maria Sklodowska-Curie National Research Institute of Oncology (MSCNRIO)—branch in Gliwice, 44-101 Gliwice, Poland; tomasz.latusek@io.gliwice.pl; 6I Radiation and Clinical Oncology Department, Maria Sklodowska-Curie National Research Institute of Oncology (MSCNRIO)—branch in Gliwice, 44-101 Gliwice, Poland; Tomasz.Rutkowski@io.gliwice.pl; 7Department of Pediatrics, Oncology and Hematology, Medical University of Lodz, 91-738 Lodz, Poland; krystyna.wyka@umed.lodz.pl; 8Department of Radiation Oncology, Dana-Farber Cancer Institute, Brigham and Women’s Hospital and Harvard Medical School, Boston, MA 02115, USA; Jonathan_Schoenfeld@dfci.harvard.edu; 9Department of Radiotherapy, Military Institute of Medicine, 04-349 Warsaw, Poland; jchalubinska-fendler@wim.mil.pl; 10Department of Radiation Oncology, Dana-Farber Cancer Institute, Boston, MA 02115, USA

**Keywords:** hypothyroidism, patient reported outcome measures, probability, radiation injuries, dose–response relationship, radiation, head and neck neoplasms, oropharyngeal cancer, NTCP

## Abstract

**Simple Summary:**

Hypothyroidism is a common complication of therapeutic irradiation in the neck area. Several dose-response models have been proposed to predict its’ occurrence based on clinical and radiomic features. We aimed to externally validate the results of five such models in a prospectively recruited cohort of 108 patients with oropharyngeal cancer. Two of the evaluated models, published by Rønjom et al. and by Boomsma et al., had satisfactory performance. Both models are based on mean thyroid dose and thyroid volume. Three remaining models, by Cella et al., Bakhshandeh et al. and Vogelius et al., performed significantly worse. Short-term change in the level of thyroid-stimulating hormone (TSH) after radiation therapy was not indicative of hypothyroidism development in long term. We conclude that the models by Rønjom et al. and by Boomsma et al. are feasible for long-term prediction of hypothyroidism in oropharyngeal cancer survivors treated with intensity-modulated radiation therapy.

**Abstract:**

We aimed to externally validate five normal tissue complication probability (NTCP) models for radiation-induced hypothyroidism (RIHT) in a prospectively recruited cohort of 108 patients with oropharyngeal cancer (OPC). NTCP scores were calculated using original published formulas. Plasma thyrotropin (TSH) level was additionally assessed in the short-term after RT. After a median of 28 months of follow-up, thirty one (28.7%) patients developed RIHT. Thyroid mean dose and thyroid volume were significant predictors of RIHT: odds ratio equal to 1.11 (95% CI 1.03–1.19) for mean thyroid dose and 0.87 (95%CI 0.81–0.93) for thyroid volume in univariate analyses. Two of the evaluated NTCP models, published by Rønjom et al. and by Boomsma et al., had satisfactory performance with accuracies of 0.87 (95%CI 0.79–0.93) and 0.84 (95%CI: 0.76–0.91), respectively. Three remaining models, by Cella et al., Bakhshandeh et al. and Vogelius et al., performed significantly worse, overestimating the risk of RIHT in this patient cohort. A short-term TSH level change relative to baseline was not indicative of RIHT development in the follow-up (OR 0.96, 95%CI: 0.65–1.42, *p* = 0.825). In conclusion, the models by Rønjom et al. and by Boomsma et al. demonstrated external validity and feasibility for long-term prediction of RIHT in survivors of OPC treated with Intensity-Modulated Radiation Therapy (IMRT).

## 1. Introduction

Radiation-induced hypothyroidism (RIHT) commonly develops in cancer survivors that receive radiation therapy (RT) for head and neck cancer (HNC). The median interval between RT and hypothyroidism is approximately 1.5 years, however later toxicities are also observed [1]. In a long follow-up (>10 years), more than 50% of patients experience RIHT [2,3,4].

Despite clinical benefits, the use of Intensity-Modulated Radiation Therapy (IMRT) was one of the factors that were reported to increase the incidence of RIHT [5,6] in contrast to earlier planning techniques, i.e., 3-dimensional conformal radiotherapy. In cases when thyroid is not contoured properly, not delineated at all or not taken into account during the optimization process, the steep dose gradients between tumor and surrounding normal tissues may result in potential overdosing in the areas where the gland is located [7]. This is of particular importance in cases when cervical lymph node regions, lying in the close proximity to the thyroid, are irradiated.

As the symptoms of hypothyroidism may be non-specific, appropriate thyroid hormone levels monitoring in patients who have undergone RT in the head and neck region is of great importance. Inadequate levels of thyroid hormones negatively impact the patients’ quality of life [8], their morbidity and mortality [5,9,10,11]; even subclinical hypothyroidism (elevated TSH with normal T3 and T4) contributes significantly to increased cardiovascular mortality [12]. Hypothyroidism symptoms and the risk it incurs may be reversed with thyroid hormone replacement, mandating routine follow-up of thyroid function and considering the thyroid gland as an organ at risk (OAR) during RT treatment planning. Still, for these efforts to yield optimal results, we need accurate tools for predicting radiation-induced hypothyroidism.

A recent systematic review of normal tissue complication probability (NTCP) models relevant to RT of the HNC [13] critically evaluated five dose–response models for RIHT [14,15,16,17,18]. To overcome the difficulty in comparing dose–response models from different studies, a relevance score was introduced to evaluate the relevance of NTCP estimations for the given patient population. The authors concluded that the most relevant NTCP model for RIHT is the one by Rønjom et al. [17] published in 2015 and using only thyroid mean dose and volume as predictors.

We aimed to externally validate the five available NTCP models for RIHT in an independent cohort of patients with oropharyngeal cancer (OPC) to verify which, if any, should be used to inform clinical decision making. We tested also which of the variables included in the original NTCP models were predictive of RIHT in the validation cohort and verified if recalibration would improve the model performance. Furthermore, we addressed the shortcomings of the previous studies by prospectively collecting data from three radiation oncology centers in Poland and by including an additional timepoint (shortly after RT) in the study design, to verify the hypothesis that monitoring TSH level shortly after RT may be useful to predict RIHT.

## 2. Results

### 2.1. Patient Inclusion and Outcome

In total, 195 patients met the inclusion criteria and were recruited in the study. Among these, 83 were excluded from the final analysis due to a lack of follow-up thyroid function assessment before recurrence/death or loss of contact, 3 due to baseline plasma TSH > 4 mIU/L and 1 due to known thyroid disease (Appendix A). A summary characteristics of the 108 patients included in the analysis (39 from center A, 13 from center B and 56 from center C) is shown in Table 1.

In the whole group, the median follow-up was 28 months (IQR 21–38 months). Thirty-one patients (28.7%) developed RIHT that required thyroid replacement therapy, meeting our primary endpoint; median time to RIHT development was 16 months (IQR 14–22 months). Neither RIHT frequency nor median time to RIHT development differed significantly between the three centers (*p* = 0.869 and *p* = 0.299, respectively).

### 2.2. External Validation of NTCP Models for RIHT

A comparison of the relevant variables (demographic, predictors and outcome) for the whole cohort and for the development cohort from each study is presented in Appendix A.

A summary of models’ performance in the whole validation cohort and in each center is presented in Table 2. The best performing models were those developed by Rønjom et al., characterized by a prediction accuracy of 0.87 (95%CI: 0.79–0.93), and by Boomsma et al., characterized by a prediction accuracy of 0.84 (95%CI: 0.76–0.91). All other models were characterized by high sensitivity (at least 90%) and low specificity. This was most evident for the model by Cella et al., which predicted RIHT in 106 (98.1%) patients and had 100% sensitivity and 3% specificity.

Model by Rønjom et al. had better specificity for RIHT prediction (90% vs. 81%), and the model by Boomsma et al. had better sensitivity (94% vs. 81%). While NTCP scores for these two models were also highly correlated—as expected given that both models included the same parameters—the model by Rønjom et al. predicted slightly lower RIHT development probabilities in patients with low risk of developing RIHT (with NTCP values from both models <0.3) and slightly higher probabilities in the high-risk patients (with NTCP values >0.8; Figure 1).

The two best models’ performance was similar and very good (areas under the curve (AUCs) 0.91–0.94) in patient cohorts from centers A and C and moderate in the smallest patient cohort (*n* = 13) from center B (AUC = 0.69 for both models; Figure 2A,B). Additionally, we visualized these models’ calibration performance using calibration plots (Figure 2C,D). Calibration plots for the three models with worse overall performance are presented in Appendix A. It can be noted that although the models by Rønjom et al. and by Boomsma et al. performed well in patients at high risk of developing RIHT, the predictions were underestimating this risk in patients with lower probabilities of follow-up RIHT. After dividing the patient cohort into the training and test set, we recalibrated the models’ outputs using Platt scaling [19], which, however, failed to improve model performance (Appendix A). Parameters of the logistic function describing recalibrated models are presented in Appendix A. 

We also analyzed separately the data from fourteen patients (13%) that had baseline TSH levels <0.3 mIU/L. Noteworthy, although the baseline TSH level is not included in any of the validated NTCP models, the model by Rønjom et al. correctly predicted the RIHT development in all these 14 patients (2 with RIHT in follow-up, 12 without RIHT) and the one by Boomsma et al. misclassified only a single patient (false positive RIHT prediction), while classifying the remaining 13 patients correctly.

### 2.3. Variables Associated with RIHT in the Validation Cohort

In univariate analyses, higher mean thyroid dose and lower thyroid volume were significant risk factors for RIHT development (*p* = 0.004 and *p* < 0.001, respectively); in the multivariable analysis, both these variables remained significant (*p* = 0.011 and *p* < 0.001, Table 3). No other factor, including baseline TSH or short-term change in the TSH level, was a significant predictor of RIHT.

### 2.4. Short-Term TSH Level Changes and RIHT

Finally, we aimed to assess whether short-term changes in the TSH or fT4 level are indicative of RIHT development in the long term. Although TSH levels decreased significantly in the short term after RT completion (*p* < 0.001), the short-term change in TSH was not different between patients with and without RIHT in follow-up (*p* = 0.844; Figure 3). This change was also not predictive of RIHT development in the multivariate model including dosimetric and clinical parameters (Table 3). The same was true for the short-term change in fT4 levels (*p* = 0.280 for comparison of patients with and without RIHT in the follow-up).

## 3. Discussion

In this prospective, multicenter study, we evaluated factors associated with RIHT in an independent cohort of 108 OPC patients treated with IMRT in three radiation oncology centers in Poland and externally validated five published NTCP models for this complication.

Since hypothyroidism negatively impacts the patients’ quality of life [8] and their morbidity and mortality [5,9,10,11], the establishment of clinically feasible models to predict RIHT is important to undertake preventive strategies in patients at high risk of this complication.

Previous reports suggested a dose–response relationship allowing a robust prediction of RIHT [13]. However, there was no quantitative analyses of normal tissue effects in the clinic (QUANTEC) report focusing on thyroid complications, as highlighted by the authors of a recent paper that attempted to revisit the dose constraints for head and neck OARs [20]. That is why we decided to focus on RIHT prediction in a contemporary cohort of OPC patients.

The best performing models in terms of accuracy and discriminative ability were those published by Rønjom et al. [17] and by Boomsma et al. [15]. Both are logistic regression-based with the thyroid mean dose and thyroid volume used as the only predictors; both were also developed in cohorts numbering >100 patients with HNC. Importantly, both models were characterized by a satisfactory performance in our dataset, with an accuracy of 87% and 84% for the Rønjom et al. and Boomsma et al. models respectively. Although the calibration was not perfect, especially in patients with lower risks of RIHT, recalibration using Platt scaling failed to improve the model performance, possibly due to the limited number of patients. 

All other three evaluated models, by Cella et al. [16], Bakhshandeh et al. [14] and Vogelius et al. [18], performed significantly worse (accuracies below 50%). This may be explained in part by the limited size of the patient cohorts in which they were developed (*n* = 65 for Bakhshandeh et al. and *n* = 53 for Cella et al.) and by the differences between the original and presented patient groups. While Boomsma et al. and Rønjom et al. NTCP models were developed in homogenous populations of HNC patients treated with chemo-RT, the study cohort for Cella et al. consisted of patients with Hodgkin’s lymphoma and the study by Vogelius et al. was a meta-analysis of four studies, two of which included patients with Hodgkin’s lymphoma. Given the differences in treatment for HNC and Hodgkin’s lymphoma, i.e., lower total target dose, a dose–response model developed using data for one of these conditions is unlikely to perform well for the other. This disparity might also explain why the model by Cella et al., which includes thyroid V30 as a predictor in the logistic regression formula, predicted RIHT in 98.1% of the patients in our cohort, where the median thyroid V30 was 100 (IQR 99.8–100).

A recent retrospective study of RIHT after IMRT in 360 OPC patients [21] reached similar conclusions to our study, i.e., that higher thyroid mean dose and smaller thyroid volume are both significantly associated with risk of RIHT in the multivariate analysis. The authors also recalibrated and evaluated two of the five already mentioned NTCP models, by Boomsma et al. and by Cella et al., each yielding AUC ROC of 0.72 and 0.66, respectively. They proposed that multicenter, prospective studies are needed to validate the available NTCP models and that monitoring of the TSH level shortly after RT, during the acute/subacute phase, may better account for the possibility of spontaneous recovery of subclinical RIHT that could impact the accuracy of the thyroid status assessment. Our results indicate, however, that monitoring the TSH level in the short-term after RT does not allow for accurate prediction of RIHT.

Noteworthy, the NTCP models published by Rønjom et al. [17] and by Boomsma et al. [15] have also gained highest relevance scores in a systematic review of NTCP models for HNC RT [13]. This result highlights the pertinence of the approach to evaluating dose–response models regarding the relevance of the patient material, study design, radiation therapy reporting and modeling approach. A comprehensive strategy summarizing possible solutions to cope with abovementioned issues was recently published and should be adhered to in future studies on the matter [22].

One limitation of our study is a small number of patients in center C (*n* = 13), which prevents an appropriate comparison of the models’ performance between all participating centers. The definition of the endpoint (clinical hypothyroidism), differs from the one defined in most studies that reported the validated NTCP models (clinical or subclinical hypothyroidism, based on the elevated TSH). We chose an endpoint based on the CTCAE classification because it is a robust and validated tool used both in state-of-the-art clinical trials and everyday practice. Another difference regarding the original studies is the inclusion of patients with hyperthyroidism (baseline TSH < 0.3 mIU/L; *n* = 14 patients). We chose to include these patients to allow the evaluation of models’ performance in a pragmatic, real-life clinical setting. Notably, NTCP models by Boomsma et al. and Rønjom et al. performed well also in this group, correctly predicting RIHT for all (Rønjom et al. [17]) or almost all (*n* = 13) patients (Boomsma et al. [15]).

Considering the potential long-term sequelae of hypothyroidism and the relatively high incidence of RIHT, it is important to include the thyroid gland as an organ at risk in routine treatment planning of RT. Recommendations for the constraint doses for the thyroid gland have been proposed by Rønjom et al. [23] with respect to the thyroid volume and may be incorporated in the clinical practice to limit the incidence of RIHT. Our study also indicates that multivariable predictive models based on patients with hematological malignancies in the H&N region should probably not be utilized to predict normal tissue complications in OPC patients.

The incidence of HPV-positive OPC patients is rapidly increasing [24] and treatment deintensification, in an effort to reduce toxicity while preserving high survival rates, is currently being explored in this group [25]. Establishing a robust NTCP model for RIHT could possibly allow treatment plan optimization [26] or selection of patients for emerging treatment techniques, such as proton therapy [27]. Given the fact that in some clinical situations (e.g., patients with small thyroids) gland sparing is often impossible, the biggest benefits of improved prediction may be in patient counseling and tailored surveillance strategies. According to National Comprehensive Cancer Network (NCCN) guidelines, TSH levels should be checked every 6–12 months after irradiation [28]. Reliable models could allow one to stratify the patients and assess thyroid function more often in the high-risk group, especially within the first years of follow-up, when the incidence of RIHT is the highest [29]. In this context, our study outlines the feasibility of predicting RIHT using published dose–response models and their potential utility in planning patient follow-up and selecting patients most likely to benefit from preventive strategies.

## 4. Materials and Methods

### 4.1. Patients

The study cohort involved 108 patients prospectively recruited at three oncology centers in Poland: Copernicus Regional Specialist Hospital in Łódź (primary center, denominated A), Radom Oncology Centre and Maria Sklodowska-Curie National Research Institute of Oncology, Gliwice Branch (called satellite centers B and C, respectively) between 01.05.2016 and 31.12.2018 and followed up until 02.2020; details regarding patient inclusion are presented in Appendix A. Inclusion criteria were: histologically diagnosed OPC, planned radical treatment, age >18 years and signed informed consent for the study. Exclusion criteria were: metastatic disease, known thyroid disease at baseline, elevated baseline serum TSH, history of thyroidectomy, history of radioiodine therapy, history of RT to the head and neck region and advanced chronic disease (heart failure—III/IV NYHA, renal failure—eGFR<30 mL/min/1.73 m^2^, liver failure—C or D score in Child–Pugh classification). 

Staging was performed according to the American Joint Committee on Cancer (AJCC) 7th edition staging system [30]. All patients enrolled in the study underwent qualification to radiotherapy according to the centre protocol, which is based on the standard treatment according to the National Comprehensive Cancer Network (NCCN). Target volumes and OARs were contoured according to the consensus international guidelines [31,32]. Dose limitations for normal tissues used for planning purposes were based on quantitative analyses of normal tissue effects in the clinic (QUANTEC). All patients were treated with IMRT (bilateral irradiation in all patients) using conventional fractionation—with a planned total dose of 69.96–70.0 Gy in 33–35 fractions, 5 fractions/week (daily Monday-Friday) [33]. The dose to target volume was planned according to the International Commission on Radiation Units and Measurements Reports 62 [34] and 83 [35]. Concomitant systemic treatment was allowed, including treatment with weekly platinum-based chemotherapy (cisplatin 40 mg/m^2^), every three weeks platinum (cisplatin 100 mg/m^2^) and induction chemotherapy according to the PF protocol (cisplatin 100 mg/m^2^ on day one with 5-FU 1000 mg/m^2^ administered by continuous infusion on days 1–4, every 21 days) or the TPF protocol (docetaxel 75 mg/m^2^, cisplatin 75 mg/m^2^ on day 1 with 5-FU 1000 mg/m^2^ administered by continuous infusion on days 1–4, every 21 days). Detailed description of the RT protocol either or not combined with systemic treatment for all patients is presented in Table 1.

For p16 immunohistochemistry, we used a CINtec p16^INK4a^ histology kit (DakoCytomation BV, Heverlee, Belgium) with a 70% nuclear and cytoplasmic staining cutoff [36]. Both positive and negative control specimens were included in every immunostaining run. Tobacco use was defined as having ≥ 10 pack-years of smoking history.

### 4.2. Treatment Planning and Contouring the Thyroid Gland

Treatment planning was carried out using Eclipse software (Varian Medical Systems, Inc., Palo Alto, CA, USA) version 13.6 or 15.1. Dose was calculated by means of an anisotropic analytical algorithm (AAA) with 2.5 mm grid size. The aim of the planning optimization was to cover at least 95% of the planning target volume (PTV) with 100% of the prescription dose. The main OARs considered were brain stem, spinal cord, larynx, mandible, parotid glands and esophagus. Sparing the thyroid gland was not part of the optimization objectives. For this study, thyroid glands were therefore retrospectively contoured by two experienced radiation oncologists according to the recent guidelines for OARs in the head and neck region [31].

### 4.3. Thyroid Function Assessment and Clinical Endpoint Definition

Laboratory assessments were performed at baseline (during the 7 days before the beginning of RT), shortly after RT (during the 7 days since the last fraction) and at follow-up (median time to follow-up 28 months). The specimens were collected during standard assessments associated with the RT treatment and follow-up visits. The normal range for TSH was defined as 0.3–4 mIU/L and the normal range for fT4 was defined as 7–22 pg/mL. To avoid bias, laboratory assessment, contouring and statistical analysis were performed by independent researchers and the person responsible for the outcome assessments was blinded to the results of the NTCP models being validated.

We defined the primary endpoint as grade ≥2 RIHT per the Common Terminology Criteria for Adverse Events grading system, version 4.03 [37].

### 4.4. Sample Size and Missing Data

Since no generally accepted approaches exist to calculate the sample size for validation studies [38], we did not perform formal sample size calculations. Instead, to maximize the power and generalizability of the comparison between evaluated models, we utilized data from all patients that met the inclusion criteria during the recruitment period and for whom follow-up information was available. We analyzed only complete cases in whom the outcome was known, i.e., when the patient was unavailable for the follow-up assessment we excluded them entirely from analysis; no data imputation was used.

### 4.5. Evaluated RIHT NTCP Models

The models by Rønjom et al. [17], Boomsma et al. [15], Cella et al. [16] and Vogelius et al. [18] were logistic regression-based while the model by Bakhshandeh et al. [14] was Lyman equivalent uniform dose (EUD) mean dose-based; for details please see Appendix B.

### 4.6. Statistical Analysis

Univariate analysis of RIHT association with an outcome was conducted by creating logistic regression models with a single predictor; odds ratios (ORs) with 95% confidence intervals (95%CI) and *p* values were reported. The following variables were considered for multivariate analysis: age, sex, surgery, tobacco use, thyroid volume, minimum dose to thyroid, mean dose to thyroid, maximum dose to thyroid, percentage of thyroid gland volume receiving no more than 30 Gy, TSH level at baseline and short-term change in TSH level. Stepwise regression with backward feature elimination was implemented, with threshold *p* = 0.1 for feature elimination. Laboratory parameters from before and after RT were compared using Wilcoxon rank sum test. Differences between RIHT and no-RIHT groups were evaluated using Mann–Whitney U test.

NTCP predictions were calculated using the original formulas reported for each model [14,15,16,17,18]. Cutoff NTCP = 0.5, representing a 50% predicted probability of RIHT development, was used for outcome prediction according to each respective model, i.e., patients with an NTCP score of 0.5 or above were defined as predicted to develop RIHT. The discriminative performance was assessed by calculating the area under the receiver operating characteristic curve (AUC ROC), Nagelkerke R2, Brier score (mean squared difference between the predicted outcome probabilities and the actual outcomes) [39], the accuracy, sensitivity, specificity and discrimination slope (difference of mean NTCP scores between patients with different outcomes) for predictions at threshold *p* = 0.5. The calibration performance of the NTCP models was assessed by calibration plots, depicting grouped observed outcome frequencies with 95% confidence intervals (95%CI) vs. mean predicted probabilities [39]. Smoothing function used to estimate observed outcome probability in relation to predicted probability was created using the loess algorithm. Correlation between NTCP scores from different models was assessed with Pearson correlation test.

Platt calibration was used to recalibrate the model predictions [19]. In this approach, a logistic function is fitted using estimates produced by the original models. The patient cohort was split in a 2:1 proportion into train and test sets. The logistic function was built using the train set and the model calibration, visualized in Appendix A and described in Appendix A, was assessed in the test set.

For all analyses, two-sided tests were used and *p* values <0.05 were considered statistically significant. Statistical analysis was conducted using STATISTICA 13.1 (TIBCO Software, Palo Alto, CA, USA) and R version 3.6.3 (2020-02-29), packages: pROC (version 1.16.2, Company, City, Country) [40,41] and CalibrationCurves (version 0.1.2, Company, City, Country) [39,41].

The analyses and reporting were done in accordance with the TRIPOD (Transparent reporting of a multivariable prediction model for individual prognosis or diagnosis) statement [38] (Appendix A). Study data, including clinical and dosimetric data used to validate the NTCP models, are available in Appendix A.

### 4.7. Ethics

The study was approved by the Bioethics Committee of the Medical University of Lodz (KE/7/10, RNN/65/18).

## 5. Conclusions

Our results from a prospectively evaluated multicenter cohort of patients with OPC confirmed that low thyroid volume and high mean dose to the thyroid gland were both significant risk factors for RIHT. We showed that the models by Rønjom et al. [17] and by Boomsma et al. [15] reliably described the dose–response relationship for RIHT and can be used in the clinic; however, owing to the described calibration issues, the predictions should be taken with caution in patients with lower predicted risk of RIHT.

## Figures and Tables

**Figure 1 cancers-12-02716-f001:**
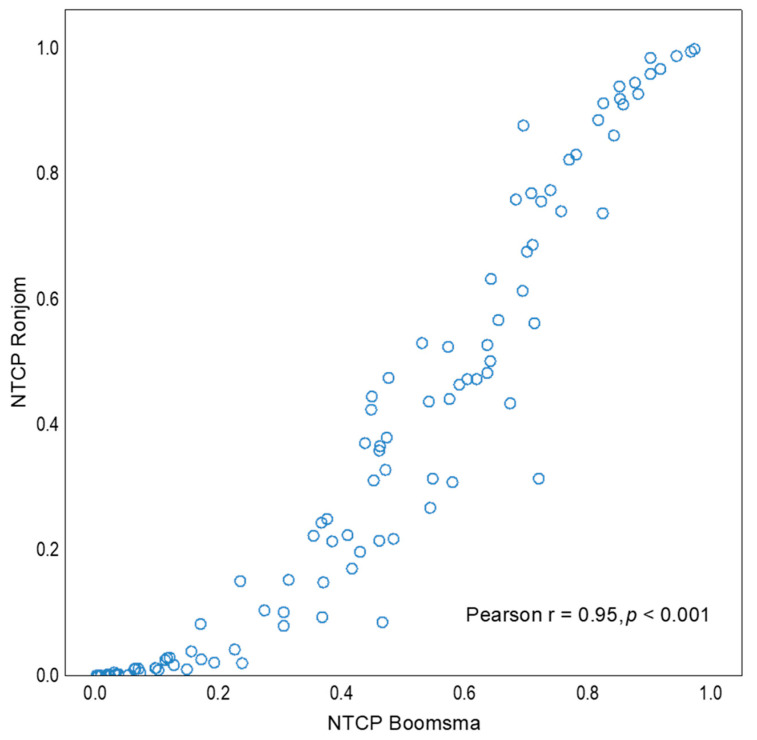
Relationship between normal-tissue complication probability (NTCP) scores predicted by models by Rønjom et al. and by Boomsma et al. for the whole study group.

**Figure 2 cancers-12-02716-f002:**
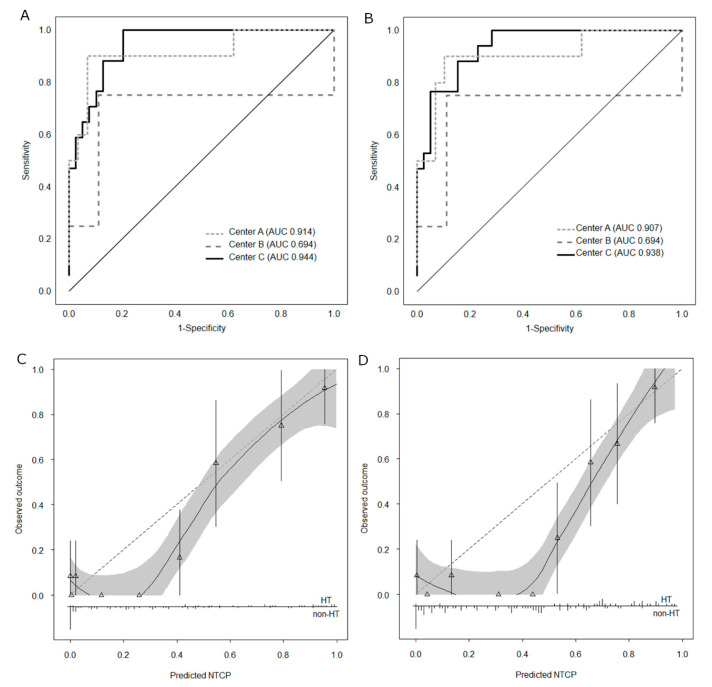
Validation of radiation-induced hypothyroidism (RIHT) predictions using the model by Rønjom et al. (left) and Boomsma et al. (right). (**A**) Receiver operating characteristic curves for each center for the model by Rønjom et al.; areas under the curve given in parentheses. (**B**) Receiver operating characteristic curves for each center for the model by Boomsma et al. (**C**) Calibration plot for the model by Rønjom et al. for all patients combined. Predictions for a perfect model would be close to the dashed reference line. The triangles represent grouped predicted probabilities (NTCP scores) vs. grouped observed frequencies of RIHT; vertical lines represent 95% confidence intervals. The loess smoothing algorithm was used to estimate the observed RIHT probabilities in relation to predicted probabilities; this function is displayed with a solid black line. The distribution of predictions in patients with/without hypothyroidism (HT) is plotted at the bottom of the graph. (**D**) Calibration plot for the model by Boomsma et al. for all patients combined. Abbreviations: NTCP = normal-tissue complication probability; AUC = area under the curve; HT = hypothyroidism RIHT = radiation-induced hypothyroidism.

**Figure 3 cancers-12-02716-f003:**
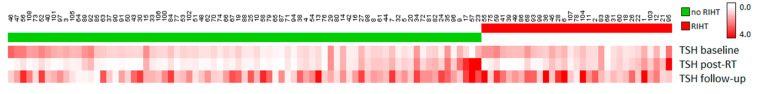
TSH levels in patients with and without follow-up RIHT (all patients with RIHT during hormone replacement therapy at follow-up). Abbreviations: TSH = thyroid stimulating hormone; RIHT = radiation-induced hypothyroidism.

**Table 1 cancers-12-02716-t001:** Patient, disease and treatment information.

Variable	Title	Center A	Center B	Center C	Whole Group
Sex	Male	25 (64.1%)	11 (84.6%)	48 (85.7%)	84 (77.8%)
	Female	14 (35.9%)	2 (15.4%)	8 (14.3%)	24 (22.2%)
Age (years)	Median (IQR)	61 (56.5–64.5)	61 (57.0–66.0)	59 (53.0–64.5)	60 (54.0–65.0)
Induction chemotherapy	Yes	4 (10.3%)	3 (23.1%)	20 (35.7%)	27 (25.0%)
	No	35 (89.7%)	10 (76.9%)	36 (64.3%)	81 (75.0%)
Concomitant chemotherapy	Yes	26 (66.7%)	11 (84.6%)	28 (50%)	65 (60.2%)
	No	13 (33.3%)	2 (15.4%)	28 (50%)	43 (39.8%)
HPV16	Positive	17 (43.6%)	4 (30.8%)	9 (5.6%)	30 (27.8%)
	Negative	22 (56.4%)	9 (69.2%)	19 (33.9%)	50 (46.3%)
	Unknown	0 (0.0%)	0 (0%)	28 (50%)	28 (25.9%)
Stage	I+II	8 (20.5%)	2 (16.7%)	17 (30.4%)	27 (25.0%)
	III+IV	31 (79.5%)	10 (83.3%)	39 (69.6%)	81 (75.0%)
Tobacco use	Yes	25 (64.1%)	11 (84.6%)	47 (83.9%)	83 (76.9%)
	No	14 (35.9%)	2 (15.4%)	9 (16.1%)	25 (23.1%)
Surgery	Yes	0 (0.0%)	0 (0.0%)	6 (10.7%)	6 (5.6%)
	No	39 (100.0%)	13 (100.0%)	50 (89.3%)	102 (94.4%)
Subsite	Tonsil	28 (71.8%)	8 (61.5%)	25 (44.6%)	61 (56.5%)
	Base of tongue	5 (12.8%)	2 (15.4%)	16 (28.6%)	23 (21.3%)
	Soft palate	3 (7.7%)	0 (0.0%)	9 (16.1%)	12 (11.1%)
	Other	3 (7.7%)	3 (23.1%)	6 (10.7%)	12 (11.1%)
Time to follow-up (months)	Median (IQR)	27.0 (21.0–35.0)	22.0 (19.0–23.0)	33.5 (24.0–41.0)	28.0 (21.0–38.0)
Mean thyroid dose (Gy)	Median (IQR)	55.2 (52.1–56.9)	55.2 (53.5–56.5)	49.2 (45.4–54)	52.7 (47.2–56.2)
Thyroid volume (cm^3^)	Median (IQR)	20.0 (14.3–30.6)	26.5 (15.5–37.4)	16.8 (12.0-22.0)	19.0 (12.9–28.2)
Thyroid V30	Median (IQR)	100.0 (100.0–100.0)	100.0 (100.0–100.0)	100.0 (99.3–100.0)	100.0 (99.8–100.0)
Mean pituitary dose (Gy)	Median (IQR)	3.9 (3.0–4.6)	3.8 (3.2–4.1)	3.8 (3.0-4.7)	3.8 (3.0–4.7)
Baseline TSH (mIU/L)	Median (IQR)	0.7 (0.3–1.1)	0.7 (0.5–1.2)	0.7 (0.6-1.3)	0.7 (0.5–1.2)
Baseline fT4 (pg/mL)	Median (IQR)	6.5 (5.1–8)	9.1 (7.2–10.1)	7.8 (6.5–8.9)	7.2 (6.0–8.8)

Abbreviations: RT—radiation therapy; TSH—thyroid stimulating hormone; fT4—free thyroxine; HPV—Human papillomavirus; IQR—interquartile range; V30—percentage of thyroid gland volume receiving no more than 30 Gy.

**Table 2 cancers-12-02716-t002:** Performance of the original NTCP models from the literature on the whole group of patients. Accuracy, sensitivity, specificity and discrimination slope were calculated for threshold value *p* = 0.5.

Performance Measure	Bakhshandeh et al. [14]	Boomsma et al. [15]	Cella et al. [16]	Rønjom et al. [17]	Vogelius et al. [18]
Discrimination					

Accuracy_0.5_ (95% CI)	0.50 (0.40–0.60)	0.84 (0.76–0.91)	0.31 (0.22–0.40)	0.87 (0.79–0.93)	0.42 (0.32–0.52)
Sensitivity_0.5_ (95% CI)	0.90 (0.74–0.98)	0.94 (0.79–1.00)	1.00 (0.89–1.00)	0.81 (0.63–0.93)	0.94 (0.79–0.99)
Specificity_0.5_ (95% CI)	0.34 (0.23–0.45)	0.81 (0.70–0.89)	0.03 (0.00–0.09)	0.90 (0.81–0.95)	0.21 (0.12–0.32)
ROC AUC (95% CI)	0.67 (0.55–0.79)	0.90 (0.82–0.98)	0.87 (0.78–0.96)	0.91 (0.84–0.98)	0.67 (0.55–0.79)
Discrimination slope_0.5_	0.101	0.427	0.025	0.528	0.120
Brier score	0.268	0.139	0.694	0.106	0.337
Nagelkerke R^2^	0.122	0.564	0.015	0.609	0.124

Abbreviations: NTCP = normal-tissue complication probability; AUC = area under the curve; CI = confidence interval.

**Table 3 cancers-12-02716-t003:** Results of univariate and multivariate logistic regression analyzing the predictor association with RIHT.

Variable	OR	95%CI
Univariate analysis		
Mean thyroid dose (Gy)	1.11	1.03–1.19
Thyroid volume (cm^3^)	0.87	0.81–0.93
Baseline TSH	1.63	0.77–3.42
TSH change pre-post RT	0.96	0.65–1.42
Age (years)	0.99	0.94–1.04
Surgery (yes vs. no)	1.26	0.22–7.25
Sex (Female vs. Male)	2.14	0.83–5.54
HPV16 (positive vs. negative)	0.85	0.31–2.33
Tobacco use (yes vs. no)	0.81	0.21–2.14
Multivariate analysis—all effects		
Mean thyroid dose (Gy)	1.11	1.02–1.21
Thyroid volume (cm^3^)	0.86	0.79–0.93
Baseline TSH	0.93	0.34–2.54
TSH change pre-post RT	0.89	0.54–1.47
Age (years)	0.98	0.91–1.05
Surgery (yes vs. no)	1.42	0.43–4.68
Sex (Female vs. Male)	1.33	0.38–4.71
Tobacco use (yes vs. no)	1.50	0.2–11.27
Multivariate analysis—stepwise regression		
Mean thyroid dose (Gy)	1.11	1.02–1.21
Thyroid volume (cm^3^)	0.86	0.80–0.93
	OR	95%CI

Abbreviations: RT—radiation therapy; TSH—thyroid stimulating hormone; HPV—Human papillomavirus; OR—odds ratio; CI—confidence interval.

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
