# Peer review of "Radiation-Induced Hypothyroidism in Patients with Oropharyngeal Cancer Treated with IMRT: Independent and External Validation of Five Normal Tissue Complication Probability Models"

_cancers, 2020, doi:10.3390/cancers12092716_

Round 1
Reviewer 1 Report
In this manuscript, the authors conducted an independent validation of five normal tissue complication probability models to predict the risk of radiation-induced hypothyroidism in a group of oropharyngeal (OPC) cancer patients. They found that 2 out of the 5 models had a satisfactory performance. These findings are important to the field because the authors independently demonstrated the validity and feasibility of using these two models to predict RIHT in OPC survivors.
Major concerns:
- The presentation of Figure 3 is confusing. It is not clear what is the key message that the authors wanted to deliver. It is also not clear how did the authors calculate the statistics.
Author Response
We thank the Reviewer for the positive feedback. We changed the presentation of TSH level differences in patients with or without RIHT in Figure 3 to make it more clear. As stated in the Methods section, differences between RIHT and no-RIHT groups were evaluated using Mann-Whitney U test.
Reviewer 2 Report
General
Risk modelling for complications is a critical topic, required for many tasks in modern radiation oncology. Consequently, many NTCP models are published. However, few investigators make the effort of validating them. As such, I would like to give my compliments to the authors for making the effort of performing a prospective trial solely aimed at this!
I am, however, quite critical about the analyses they performed. Validating models in a way that indeed provides the field with new knowledge that can be used to change practice is a big task that requires considerably more work than currently included in the manuscript. See detailed comments for details.
Specific
Page 2, line 70-75: If I understand correctly, the authors pursue 3 aims: model validation, model updating and extending it.
Page 3, line 92-95: The multivariate analysis performed here is insufficient to conclude that no other variable plays a role. Weak predictors not significantly associated univariably may actually predict residual variance as second or third variable in a multivariable analysis. The authors do need to revise their modelling strategy by not excluding variables for the sole reason of lack of univariate significance. In addition, to develop understanding of the reason why parameters are not selected, to evaluate various potential reasons for non-selection. (Not being predictive is only one possibility next to collinearity and other potential reasons.)
Page 4, line 104-106: Though often reported, the area under the ROC curve is not very informative of practical applicability of the model and should not be used as a primary performance metric to judge the value of a model.
Page 4, line 106-110: I’m assuming that this is the only part where the NTCP=0.5 thresholding was used? Also, this analysis is not very useful: It represents one point on the ROC curve, which is therefore more informative.
Page 4, line 114: By itself this correlation doesn’t have many implications, besides the fact that this is expected because the models contain the same risk factors, with any deviation from r^2=1 resulting from differences in calibration. Please leave it out.
Page 5, line 118-121: The aim of a risk model is … to estimate risk. The primary quality should be in its calibration. What’s really missing in the validation analysis is calibration plots of the 3 other models indicating how well they do this in this external validation setting. Even though the Brier score does this to some extent, it’s hampered by quite a few limitations. The underlying calibration plots are much more informative. Please include a multi-panel figure with one for each model. In addition, when looking at figure 2c and d, these models actually suffer badly from systematic deviations, especially towards the lower risks. Since one of the applications of these models is to estimate the benefit of and optimize use of new (and more precise) technology, this is where deviations really hurt. Investigating possible reasons would be really valuable. (See also next remark.)
Results missing: An analysis of reasons for discrepancies between model and validation data is missing entirely. The authors should perform analysis to investigate possible reasons. A very common reason for models to be inconsistent with external validation data is a shift in case mixes, which can be explored by testing population characteristics of the development set against their validation set. This would provide a more solid basis for discussion on the interpretation of the results of the validation analysis. Other possible reasons should be explored in the results before discussing them in the discussion as well.
Results, missing: Validation of a model consists of two parts: One is to test for differences between validation data and a previously developed model. If differences are found, it remains important to know whether this can be solved by recalibration to obtain insight into the relevance of the risk factors for risk estimation.
Page 7, line 144-153: Similar to my remark on the use of univariable analysis as a preselection for constructing a multivariable model, this analysis is flawed. Part of the variability will be explained by patient/treatment-related factors. Once corrected for that, in multivariable analysis short-term changes may still significantly explain residual variance in the outcome.
Line 165-166 (but also the subsequent lines): See previous remark on criteria for judging model performance and recalibration. This conclusion is to simplistic and insufficiently supported by the analysis.
Line 173-175: Demonstrating this requires additional results. (See one of the previous remarks.)
Line 192-193: As indicated previously, I wouldn’t give up on this yet :-). At least the analyses need to be improved in order to better substantiate this. Moreover, no single variable will allow accurate prediction. It’s a multivariable problem and the question is rather whether a variable can improve prediction.
Line 200-205: A change in endpoint will usually cause calibration issues. As such, the poor calibration in figs 2c and 2d is not surprising and recalibration is indeed indicated in order to judge the utility of the variables for risk estimation.
Line 213-214: This is not related to and not supported by the analysis perfomed by the authors. Remove, or provide evidence for a parallel nature. (Note that mean dose as a predictor is not evidence, unless a comprehensive analysis of other dose metrics is performed.)
Line 222-224: All these applications require calibration performance, rather than classification performance.
Line 324-330: Given the above remarks, none of the conclusions is currently supported. However, revising as suggested above might repair this, or lead to alternative (equally interesting) conclusions.
Round 2
Reviewer 1 Report
My concerns are addressed in the revised manuscript.